# Confessional Instruction or Religious Education: Attitudes of Female Students at the Teacher Education Faculties in Serbia

Uroš V. Šuvaković [1], Jelena R. Petrović [2,*] and Ivko A. Nikolić [1]

1    Teacher Education Faculty, University of Belgrade, 11000 Belgrade, Serbia
2    Military Academy, University of Defence, 11000 Belgrade, Serbia
*    Correspondence: jelenailicpetrovic@gmail.com

**Abstract:** This paper presents an empirically study on the attitudes of female university students at all Teacher Education Faculties in Serbia (TEFS). For the purposes of this study, a survey was prepared to be completed by students online, and virtual exponential non-discriminative snowball sampling was applied. The independent variables were religion, major subjects, year of study, age, the completion of secondary schooling, whether an optional subject was studied during previous schooling, and whether female students were employed. The dependent variables were the respondents' attitudes to religious instruction and civic education. The sample included 372 students from all TEFS, and the research was conducted in the period from 15 May to 8 June 2022. The data were processed with the aid of nonparametric statistics. The results showed that religion did not contribute to differences in students' attitudes and opinions regarding the method of performing religious instruction and civic education, but that some other factors contributed to it, such as previous experience with these subjects and whether the respondent had completed secondary schooling. These results may be interpreted in the light of the weaknesses of the confessional model, that is, the lack of knowledge regarding the basic paradigms of other confessions, which is of great importance for countries such as Serbia which have numerous national minorities and religious communities.

**Keywords:** religious education; confessional instruction; civic education; university students; Serbia

## 1. Introduction

Religion is one of the oldest spiritual creations, but neither Christianity nor Islam has succeeded in achieving planetary acceptance, although they indisputably strive towards it. In addition, religious practice is not the same everywhere in the world, not even within the same confessional group. This is commonly observed by foreigner travelers in Serbia. For instance, Rodolphe Archibald Reiss observed that Serbs were not religious, that they did not believe in God "as shown in the Bible", but had transformed their religion in line with folk customs, thus creating the "people's church or, rather, national tradition", in which national heroes and saints were mixed for the purpose of preserving patriotism, which was also developed by priests who were not "church people but ardent patriots with all their faults and virtues" (Reiss [1928] 2019, p. 299). A similar conclusion has also been reached by modern researchers, who have pointed that the religion among Serbs "cannot be called Christian, at least not in the literal meaning of the word. Serbs did not abandon their pagan faith and replaced it with a new one . . . They accepted Christianity, but often understood it in a non-Christian manner" (Bandić 2004, pp. 11–12). This is why some sociologists of religion coined the term "*four-rite believers*" for Serbian adherents of Orthodox Christianity (characterized by christening, patron saint's days, church weddings and funerals with a religious service), emphasizing at the beginning of the 21st century that "Serbs are religious in a manner of traditional belonging without believing" (Đorđević 2009, p. 62). In any case, the observance of patron saint's days in the form of feasts in honor of the patron stain of the home, school, institution, profession or enterprise, as well as the

settlement on dates dedicated to that particular patron saint in the church calendar of the Serbian Orthodox Church, exist only among Serbs[1], and these practices are thought to be a remnant of pre-Christian beliefs and a result of the Christianization of folk customs for the sake of strengthening the influence of the church among the people (Ivanović Barišić 2008), i.e., marking the day when a Slavic or Serbian family accepted Christianity (Vukomanović 2001, p. 71), although there are other interpretations of their origin as well (Kalezić 2000; Pavković 2015).

Religion is a particular view of the world characterizing a small or a large social group (see Figure 1). In that respect, it is different from science, which has always been founded on universality (Jovanović and Šuvaković 2014). "While beliefs of the members of sects and various groups are based on personal choice or temperament and while it divides people, the scientific procedure unites people" (Cohen and Nagel 2004, p. 411).

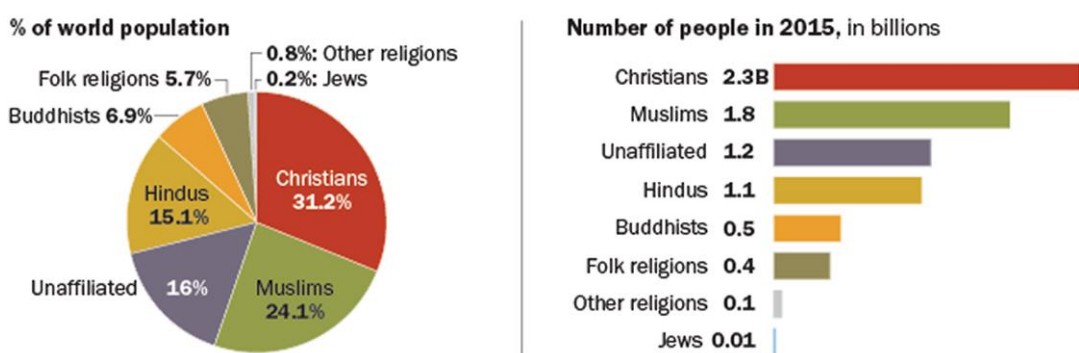

**Figure 1.** Global religious populations. Source: Hackett and McClendon (2017).

As in other countries in the real socialist bloc, religion was not welcome in socialist Yugoslavia. Nevertheless, at that time, the attitudes of the authorities towards the church and religiousness differed among the former Yugoslav states: in the Catholic parts (Slovenia, Croatia), they were tolerated, whereas in Bosnia and Herzegovina, Montenegro and Serbia, they were more or less restrained and in Macedonia they were favored[2]. The fall of the Berlin Wall and the collapse of real socialism, which was largely influenced by the Roman Catholic Church and Pope John Paul II in person, who had first served as Cardinal Wojtyla in Poland and openly supported the anti-communist "Solidarity" movement (Wałęsa and Rybicki 1992), led to the revival of the influence of the church and religion in the post-socialist territories in general (Berger 1999, p. 6; Kulska 2020; Ramet 2014), including post-Yugoslav territory as well (Blagojević 2009; Petrović 2011; Petrović and Šuvaković 2013; Raduški 2018, pp. 28–33; Smrke and Uhan 2012; Vukomanović 2016). However, this revival did not occur to an equivalent extent in all these territories (Bogomilova 2003, pp. 517–20; Flere and Klanjšek 2008)[3] or in the same manner (Filipsone 2005; Kulska 2021).

In the study of religious instruction, it is generally possible to distinguish between three European models: *secular*, which implies that no form of religious instruction is organized in state-owned schools; *confessional*, which implies that schools organize confessional instruction (religious teaching) for students who belong to a certain confessional group, with the modalities of compulsory or optional subjects, and *religious-educational*, within which religion is studied as a cultural, social phenomenon (e.g., through subjects related to modern world religions, dealing with sociology, philosophy, ethics and the history of religion as a social and cultural phenomenon (Maksimović 1998; Teece 2010, p. 101), whereas in

non-European countries we can also find theocracies in which *the whole educational system is permeated by a certain religion* "as integral part of education" (Ivanović 2015, p. 15).

The experience in the post-Yugoslav period related to the introduction of religious instruction has been diverse. In the last decade of the 20th century, the confessional model of religious instruction was introduced in Croatia and Bosnia and Herzegovina, whereas Slovenia applied the non-confessional religious-educational model, dealing with different religions from the perspective of ethics (Kuburić and Moe 2006). In Montenegro, religious instruction has not been introduced in state-owned schools to date, whereas in Macedonia, the confessional model of religious instruction was introduced and then, after one year, declared unconstitutional and abolished, whereas, as of September 2010, students in the higher grades of primary schools were given the choice of studying one of three optional subjects (Introduction to Religions, Classical Culture in European Civilization and Ethics in Religion) (Koceva 2015). In this way, Macedonia opted for the non-confessional model of religious education. In Serbia, despite the increasingly strong influence of the Serbian Orthodox Church from the end of the 1980s, there was no religious instruction in state-owned schools until the political changes of October 2000. This subject was introduced as a confessional form of religious instruction in all primary and secondary schools in Serbia in November 2001 in line with the decision of Prime Minister Zoran Đinđić, which was promoted by the Serbian Orthodox Church because it was supported by other traditional religious communities in Serbia as well (the Roman Catholic Church and the Islamic community being the largest ones after the Serbian Orthodox Church, by the number of believers)[4] "as a significant correction in the system of social values" (Bazić 2011b, p. 221). Civic education was introduced as an alternative subject (Šuvaković 2014, pp. 171–72). This was followed by the re-establishment of the Faculty of Orthodox Theology within the University of Belgrade (2004). Ideologically speaking, in the reform initiated at that time, "the teaching content of Yugoslav and social heritage were omitted from the educational programs, while religious, spiritual content and content of neoliberal ideology of cultural patterns of Western capitalist societies were introduced" (Bazić and Sekulić 2017, p. 68). Traditional churches and religious communities were legally guaranteed the right to choose "their" religious teachers, in line with criteria that were not of secular character, which also applied to the professors of the Faculty of Orthodox Theology. However, the work status of religious instructors has remained unresolved to date, with these instructors signing a new one-year agreement for every school year. In the case that these instructors are not included in the list of their church or religious community, they will lose their jobs.[5] Questions regarding the textbooks used for this subject are also unresolved. Before 2016, when the curriculum and syllabus were changed, there had been textbooks of Orthodox catechesis covering the entire primary and secondary school curriculum, written by an Orthodox Bishop of Braničevo, Ignatije (Midić), and published by the Serbian State Textbooks Publishing, Belgrade (Ivković 2017, pp. 167–68)[6]. Subsequently, the Serbian Orthodox Church published textbooks for this subject. There was a similar situation regarding textbooks for the students of Roman Catholic confessional studies and of Islamic faith. Textbooks for these subjects are not subject to assessments by relevant expert state bodies, but decisions regarding them are made by local religious communities, which also have a say in relation to school curricula and syllabuses for these subjects. As for civic education, the situation also remains unresolved. The subject is taught by teachers without a sufficient number of classes within their own disciplines, and textbooks have not been written yet. Workbooks are available from various publishers, as well as manuals for teachers published by the relevant Minister of Education or by a specialized state agency that deals with the development of quality in education (ZUOV).

More than two decades since the introduction of religious instruction and civic education in primary and secondary schools in Serbia, we are interested in the attitudes of the students at TEFS regarding the nature of these subjects and the applied model of religious instruction. Teacher education faculties were chosen because, upon the completion of their studies, these students will, by nature of their profession, represent children's first

encounter with society, in the form of schooling. Therefore, they function as role models and behavioral models for schoolchildren and, more than any other teaching professionals, religious instructors attempt to influence children's socialization.

Considering that the introduction of religious instruction was a result of democratic changes in Serbia, and that a consensus has been reached among the traditional churches and religious communities in Serbia regarding its confessional model, our first hypothesis was as follows:

Female students in teacher education faculties would have a positive attitude towards the current status and model of religious instruction in Serbia.

Taking into account the fact that civic education as an alternative subject was introduced in an ad hoc manner, and that even two decades later there are no teaching staff that have been specifically trained in this field and no adequate textbooks written for this subject, our second hypothesis was as follows:

Female students in teacher education faculties would think that the subject of civic education as an alternative subject to religious instruction should be devised in a better manner.

## 2. Materials and Methods

Sample: virtual exponential non-discriminative snowball sampling was applied (Parker et al. 2019). The sample included 372 female students of TEFS, aged 19–40 years. In Serbia, the teaching profession is predominantly female, and that is why male respondents were excluded from the study (there are fewer than 9% male students in the total student population from teacher education faculties in Serbia (RSZ 2021)). In addition, TEFS are organized in such a manner that they are divided into two departments: pre-school teachers and school teachers. There are a total of seven TEFS, the largest of which is the one in Belgrade, with two branches in the province, whereas the others are evenly distributed throughout the country. This study included 211 students majoring as teachers and 161 majoring as pre-school teachers. In relation to their national affiliation, 77% of students were Serbian and 16% were Bosniak, whereas the remaining 7% belonged to other nationalities (cf. Figure 2).

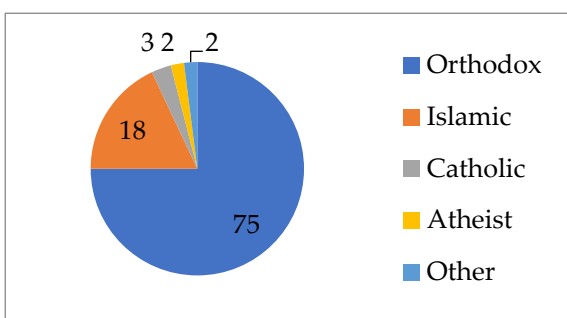

**Figure 2.** Students' religious affiliations (%).

Procedure: The data were collected via a survey that was made available online to the students and the completion of which was voluntary and anonymous. A link to the survey questionnaire (see Appendix A) was forwarded to the students of TEFS through social networks, student social groups, faculty websites, etc. Several dozen access-points for the questionnaire were provided.

Tools: The tool used in the research was an online survey questionnaire constructed for the purpose of the study, which included questions regarding the respondents' basic data, such as questions regarding their attitudes towards religious instruction and civic education.

Variables: The independent variables in this study were: major subjects, year of study, age, the type of the secondary school completed (grammar school or secondary vocational school), the subject taken by the students in their previous schooling (civic education or religious instruction) and the students' employment status.

The dependent variables were: the respondents' attitudes towards religious instruction, the respondents' attitudes towards civic education, their attitudes towards the status of religious instruction, their attitudes towards the status of civic education, and their opinions regarding how religious instruction should be organized in state-owned schools in Serbia.

### 3. Results

Among the surveyed female students, 27% graduated from a grammar school, whereas 73% graduated from some other type of secondary school. Twenty-three percent of the female students were employed. During their previous school cycle, 22% female students attended civic education classes, whereas 78% students attended religious instruction classes.

Considering the total sample, we found that the female students had higher opinions regarding the methods used to provide religious education as compared to those used to provide civic education. However, it should be observed that as many as 44% students thought that the system should be modified in relation to the subject of civic education, whereas 30% of them held this opinion in relation to religious education (Figure 3).

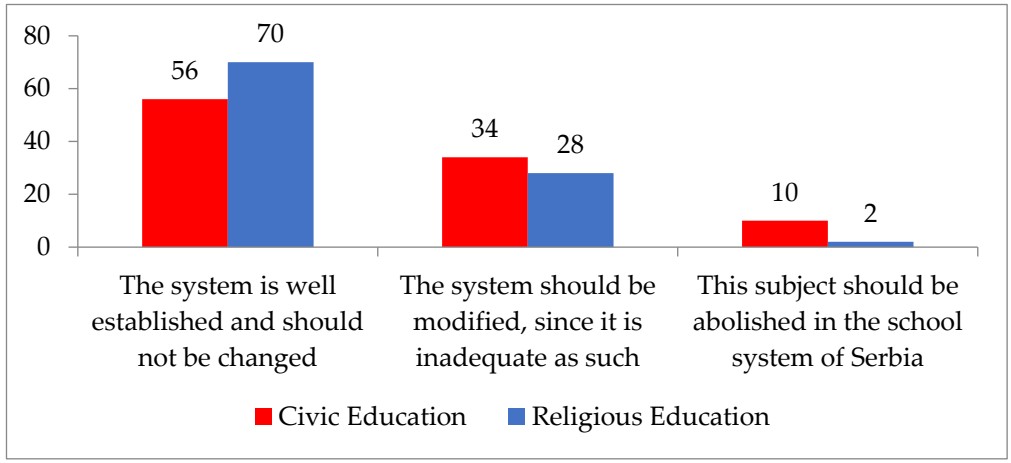

**Figure 3.** Comparison of the students' attitudes towards the quality of civic education and religious education classes (%).

Such modifications concerned religious (confessional) instruction becoming a compulsory subject, which was advocated by as many as 40% of respondents, whereas slightly more than one half of them (57%) believed that religious education should retain the same status as it had in Serbia's school system at the time of the study—that of an optional subject. A negligible proportion of the respondents were in favor of abolishing the subject of religious education, whereas as many as 10% of the respondents were in favor of abolishing civic education, and 70% of them were in favor of its maintaining its status as an optional subject, whereas 20% believe that the subject should be compulsory (Figure 4).

The idea of the secular model of teaching was considered acceptable by only 6% of the respondents. The students mainly (62%) favored the confessional instruction approach used in the (current) confessional model and regarding the notion that lower grades of the first schooling cycle should be taught according to the confessional model and higher grades of religious education should be taught according to the religious-educational (non-confessional) model, they did not tend to make a distinction. Slightly fewer than 1/4 of the respondents were in favor of the non-confessional model of teaching (Figure 5).

With further stratification of the sample and the introduction of the remaining independent variables, certain differences were found that proved to be statistically significant. Regarding the respondents' ages and the quality of religious instruction, we observed that most second- and third-year students thought that the system was good and should not be changed ($\chi^2$ = 129.390, df = 6, *p* = 0.000). However, most younger students thought that religious instruction should be an optional subject ($\chi^2$ = 198.359, df = 6, *p* = 0.003). Among

the student population there was obviously a positive attitude towards religious instruction of this type and that respondents generally thought that potential changes should not be made in regard to the format in which this subject was taught.

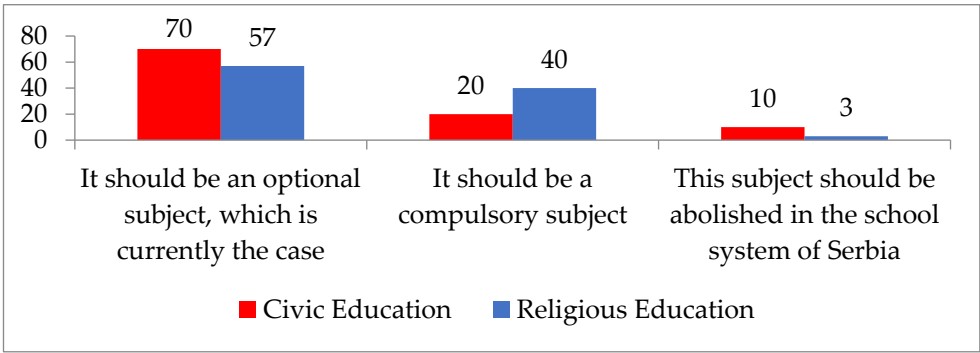

**Figure 4.** Comparison of the students' attitudes towards the status of civic education and religious education classes in the school system of the Republic of Serbia (%).

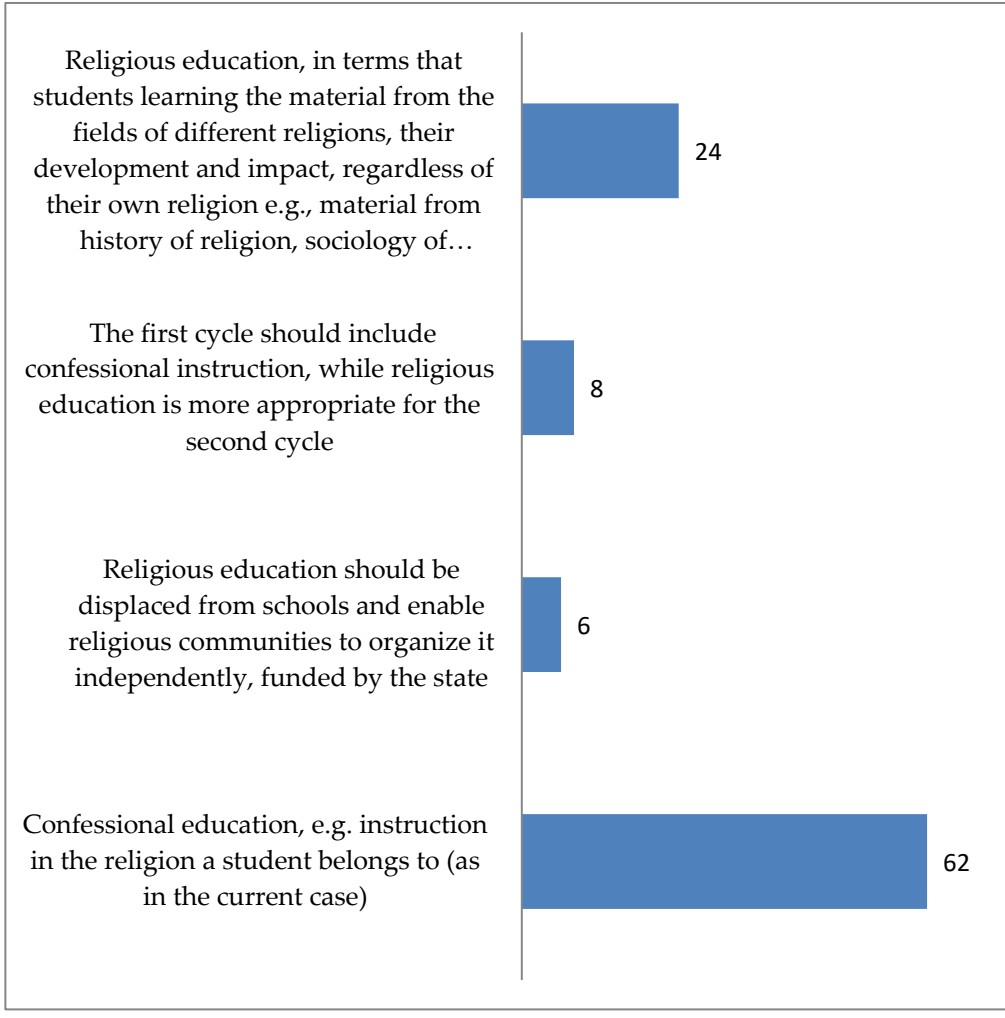

**Figure 5.** Students' opinions about the manner in which religious education was organized in primary schools in Serbia (%).

Regarding the respondents' previous secondary school education, most students who had graduated from secondary schools other than grammar schools thought that the system regarding religious instruction was good and should not be changed ($\chi^2$ = 189.347, df = 6, $p$ = 0.000). Statistically significant differences were also observed regarding whether the respondents attended religious instruction or civic education classes. The largest number of the students who attended religious instruction classes thought that the system is good and should not be changed ($\chi^2$ = 45.563, df = 6, $p$ = 0.000). Moreover, the largest number of students who attended religious instruction classes thought that religious instruction should be a compulsory subject ($\chi^2$ = 31.497, df = 14, $p$ = 0.005) and that civic education should be abolished ($\chi^2$ = 18.121, df = 6, $p$ = 0.006).

Statistically significant differences were also observed in relation to employment ($\chi^2$ = 41.464, df = 6, $p$ = 0.024). The largest number of the students without employment thought that the religious instruction system was well established and should not be changed.

## 4. Discussion

Female students of TEFS generally had a more positive attitude towards religious instruction than towards civic education. That attitude was reflected in the opinion of the majority that the existing system was good and that the model should be maintained, as well as in the agreement with the thesis that in the future it is necessary to consider introducing this subject as a compulsory one, using the confessional model (Figure 5). Most respondents were not aware of the need to become familiar with the teachings of other churches and religious communities, which is problematic in a study of future teachers and pre-school teachers in a multiregisious, multiconfessional and multiethnic country such as Serbia. On the other hand, there was far less satisfaction with the current concept and organization of the subject of civic education with regard to the previous two parameters (Figures 3 and 4).

The results also showed that there were certain differences in relation to individual variables. Young students who did not attend grammar school and who were unemployed tended to a larger extent to be satisfied with the current state of affairs and reported that they would like religious education to remain an optional subject. In contrast, the respondents who attended classes in religious instruction believed that it should be compulsory, but they also expressed the attitude that civic education should be abolished, which in certain terms pointed to more extreme attitudes, as compared to those students who attended civic education classes and who did not show such a tendency.

On the other hand, the lack of affirmative results obtained regarding the respondents' attitudes towards civic education point to the necessity of modifying the content of the subject itself, since there was agreement as to whether the subject should maintain its optional status. However, it should be noted that slightly more than 3/4 of the respondents in the previous school cycle attended religious instruction classes and had no personal experience with civic education. Furthermore, practical experience may have contributed to a situation in which the students who were already employed and who had perhaps been in contact and cooperated with people with other beliefs were better able to perceive the importance of religious education in terms of acquiring knowledge about other religions and beliefs as compared to the knowledge acquired solely through the teachings of their own belief system.

## 5. Conclusions

Owing to the influence of the Serbian Orthodox Church, supported by other traditional churches and religious communities in Serbia (primarily, the Roman Catholic Church and the Islamic community), as well as the need among the new neoliberal authorities to establish better relations with these churches and religious communities, in 2001, religious instruction was introduced in all Serbian state-owned primary and secondary schools, an arrangement which lasted for 12 years. The confessional model was applied, although

it was classified as a subject of an extra or additional type, but it obtained the status of an optional subject the following year, with civic education as an alternative subject. The confessional model of religious instruction has been applied in a number of European countries. The model applied in the schools in Serbia, naturally adjusted to its religious, confessional, and ethnic specificities, is most similar to the one applied in Germany; thus, some legal experts have considered it a "legal transplant"[7] (Avramović 2016, p. 28).

However, it is debatable whether this is also the most suitable model for a multireligious, multiconfessional and multi-ethnic country such as Serbia. It might be more appropriate to teach children about the scientific view of religion, the knowledge of different world religions (Liessmann 2008, p. 55), or "how science explains the existence of religion in historical, biological, psychological and sociological terms" (Mahner and Bunge 1996). On the other hand, there is an indisputable aspiration among churches and religious communities to gain the greatest influence possible over institutional education (Bazić 2011a) for the purpose of exerting an influence on the socialization process, which is also confirmed by findings showed that the introduction of religious instruction according to the confessional model has contributed to an increasing number of people declaring themselves as believers, primarily among young people (Cvitković 2017, p. 21). Religious instruction according to the confessional model is not primarily aimed at acquiring knowledge about religion as a social phenomenon, but at developing religion and creating believers (Kuburić and Zuković 2010, p. 58). When believers stopped visiting places of worship and, as we have already pointed out in the case of Serbia, church attendance was reduced to several times per year, the church decided that religious instruction teachers should reach future believers. Therefore, the confessional model of religious instruction—in fact religious indoctrination—was introduced into the educational system of Serbia, despite serious resistance from the then-Minister of Education himself (Aleksov 2003). There was consent among the traditional churches and religious communities and willingness among the authorities in Serbia to meet such requirements. Civic education was introduced with no preparation, like a "fig leaf" designed to ensure an impression of respect for the constitutional principle of secularity, instead of constituting a seriously conceived subject that would educate young generations for democratic purposes (Maksić and Pavlović 2017; Stanković 2018; Šuvaković 2019, p. 962). Hence, a significantly more critical attitude was observed among the surveyed students towards this school subject, and there was a large share of respondents in favor of the modification of this subject, and 1/10 even supported its abolition, which fully confirmed our second hypothesis.

Twenty-one years after the introduction of this policy, by conducting a survey of the female students of TEFS—who will teach at schools and pre-school institutions upon the completion of their studies—we obtained empirical data that inevitably led us to draw conclusion about the dominant acceptance of the confessional model of religious instruction in Serbia. This confirmed our first hypothesis regarding the model of religious instruction. As for the status of the subject, it may be said that our first hypothesis was mainly confirmed, since as many as 40% students thought that religious instruction should change its status and become a compulsory subject instead of an optional one. This conservative approach, which is particularly problematic when it comes to multireligious, multiconfessional and multiethnic societies such as Serbia, was predominant primarily among those students who had previously attended religious instruction classes and who favored, in quite a large proportion, this subject becoming compulsory. The causes for such attitudes may be sought in the respondents' previous personal experience, non-critical opinions, the teachings of their previous schooling, etc., but in our opinion they are the result primarily of the influence exerted on the students through the full consent of the traditional churches and religious communities in this respect. This is also related to the post-socialist transition of Serbia, as well as the ever-present feeling of one's own collective identity being threatened, which is strengthened in this manner by a clear (religious) distinction from the characteristics of otherness (cf. Gašić Pavišić and Ševkušić 2011, p. 70), which are not learnt about by students at all. The confinement of schools within the framework of the locally dominant

confessional model is the greatest fault of this concept of religious instruction, which was not observed by the surveyed students. Althusser's famous observation that the school replaced the church as the most effective ideological state apparatus in modern capitalist societies (Althusser 1970) could be modified now, at least in the case of Serbia. Namely, the church has clutched the school system so firmly that it now performs the function of religious socialization that the church is no longer able to perform successfully on its own. The students' attitudes observed in this study show the success, from the point of view of the church, of this church–school symbiosis, whereas it is uncertain whether it will have an equally successful effect regarding the state's interests, not only due to the existence of multireligiousness, multiconfessionalism and multiethnicity, but also due to the necessity that education should rely on the scientific principle.

The limitations of the conducted research derive from the fact that only the attitudes of the students of TEFS were examined, and not of those of other faculties in Serbia. The inclusion of students from other faculties in the sample would have resulted in a substantially higher presence of male students, which was not possible in the examination of the attitudes of students of TEFS, considering the almost complete feminization of the teaching/pre-school teaching profession in Serbia. Moreover, sampling of the entire student population would require checking whether there were certain differences, and the point of selecting students from TEFS would be lost. Namely, schools are the first places in which children encounter society, and the attitudes of teachers and pre-school teachers play a very important role in that stage of children's socialization. Methodologically speaking, it is possible to think about applying a different type of sample (e.g., the stratified quota sampling approach), which would ensure the proportional presence of each of the two groups of students (those who had attended religious instruction classes and those who had attended civic education classes) in the total sample. However, this would also have its limitations in terms of restraining participation in the survey, and this is related to the social pressure of the environment (particularly in the case of smaller locations which also have faculties, as part of the rather disjointed network in Serbia). Perhaps the best way of overcoming the abovementioned limitations would be to repeat this study at teacher education faculties and to expand the surveyed sample to include the overall student population, while using the type of sampling applied in this research.

**Author Contributions:** Conceptualization, U.V.Š.; methodology, U.V.Š. and J.R.P.; formal analysis, J.R.P.; investigation, I.A.N.; writing—original draft preparation, U.V.Š., J.R.P. and I.A.N.; writing—review and editing, U.V.Š. and J.R.P.; visualization, U.V.Š.; supervision, U.V.Š.; project administration, U.V.Š. and J.R.P.; funding acquisition U.V.Š. and I.A.N. All authors have read and agreed to the published version of the manuscript.

**Funding:** This research was funded by the Ministry of Education, Science and Technological Development of the Republic of Serbia, grant number 451-03-1/2022-14/4, received by Belgrade-Teacher Education Faculty.

**Institutional Review Board Statement:** Ethical review and approval were waived for this study because the survey conducted in the study was voluntary and anonymous, the participants could not be identified when completing the online questionnaire.

**Informed Consent Statement:** Informed consent was obtained from the participants.

**Data Availability Statement:** Data presented in this study are available upon request from the corresponding author. The data is not publicly available because it was collected solely for the purpose of scientific research.

**Acknowledgments:** This research was funded by the Ministry of Education, Science and Technological Development of the Republic of Serbia, grant number 451-03-1/2022-14/4, received by Belgrade-Teacher Education Faculty.

**Conflicts of Interest:** The authors declare no conflict of interest. The funders had no role in the design of the study; in the collection, analyses, or interpretation of data; in the writing of the manuscript, or in the decision to publish the results.

### Appendix A. The Questionnaire on Behaviour and Attitudes of Students of Teacher Education and Pedagogy Faculties in Serbia in Relation to Some Subjects in Schools

*Due to the volume of the questionnaire, we provide an extract here—only those questions to which the answers were statistically processed and used in our article.*

| | |
|---|---|
| 1 | Which faculty are you studying at? |
| | Teacher Education Faculty in Belgrade |
| | Teacher Education Faculty in Belgrade—Department in New Pazar |
| | Teacher Education Faculty in Belgrade—Department in Vršac |
| | Faculty of Pedagogy in Užice |
| | Faculty of Pedagogical Sciences in Jagodina |
| | Faculty of Pedagogy in Vranje |
| | Teacher Education Faculty Prizren—Leposavić |
| | Faculty of Pedagogy in Sombor |
| 2 | What is your year of study? 1 |
| | 2 |
| | 3 |
| | 4 |
| | 5 |
| 3 | What are your majors? |
| | 1—Teacher |
| | 2—Pre-school teacher |
| | Other: |
| 4 | Gender: |
| | Male |
| | Female |
| 5 | How old are you? |
| | Select: |
| 6 | What is your nationality? |
| | This question is of a private nature and you are not obliged to answer it. |
| | Serbian |
| | Croatian |
| | Macedonian |
| | Bosnian/Muslim |
| | Yugoslav |
| | Hungarian |
| | Albanian |
| | Other: |
| 7 | What is your religion/confession? |
| | This question is of a private nature and you are not obliged to answer it. |
| | Serbian Orthodox |
| | Catholic |
| | Islam |
| | I am an atheist |
| | Other: |
| 8 | What is your previous degree of education? |
| | Grammar school |
| | Secondary vocational school |
| 9 | Are you employed? |
| | YES |
| | NO |

_______

| | |
|---|---|
| 25 | Which of these two subjects did you attend previously (during your primary and secondary school)? |
| | Religious Instruction |

Civic Education

26　What is your attitude towards Religious Education as a subject in the school system of Serbia?

The system is well-organized and it should not be changed

The system should be modified since it is currently inadequate

Religious Instruction should be abolished as a subject in the school system of Serbia

27　What status should Religious Instruction have in the school system of Serbia?

It should be a compulsory subject

It should be an optional subject, just as it is currently

Religious Instruction should be abolished in the school system of Serbia Other:

28　In your opinion, Religious Instruction in primary schools in Serbia should be organized as:

Religious Instruction and/or education about the religion a student belongs to (as is currently the case)

Religious education in which students master the material from the field of different religions, their development and impact, regardless of the religion they belong to (e.g., material from world's religions, history of religion, sociology of religion, philosophy of religion, etc.)

Religious Instruction should be displaced from schools and churches and religious communities should be able to organize it independently at the expense of the government

During the first cycle, there should be Religious Instruction, whereas in the second cycle Religious Education would be more adequate

29　What is your attitude towards Civic Education as a subject in the school system of Serbia?

The system is well-organized and it should not be changed

The system should be modified since it is currently inadequate

Civic Education should be abolished as a subject in the school system of Serbia

30　What status should Civic Education have in the school system of Serbia?

It should be a compulsory subject

It should be an optional subject, just as it is currently

Civic Education should be abolished in the school system of Serbia

Other:

## Notes

[1]　In 2014, the patron saint's days were included in UNESCO's Representative List of the Intangible Cultural Heritage of Humanity with the following explanation: "Slava, celebration of family saint patron's day, Serbia" (UNESCO 2014).

[2]　Former socialist authorities exerted pressure on the Serbian Orthodox Church to grant autocephaly to the Macedonian Orthodox Church, which was seen as a part of completing the Macedonian national identity. The Serbian Orthodox Church recognized its autonomy in 1959, whereas the Macedonian Orthodox Church, supported by the atheist authorities, declared its autocephaly in 1967 (Janjić 2018; Matevski and Matevska 2018, p. 1444). After more than half a century, in May 2022, the Serbian Orthodox Church recognized the autocephaly of the Macedonian Orthodox Church, i.e., the Ohrid Archbishopric, and re-established relations with it, which was then carried out by some other Orthodox churches (e.g., the Russian Orthodox Church).

[3]　The strength of the Church in Montenegro was crucial for the occurrence of the last clerical revolution in Europe 2019/2020, when mass processions led by the archbishops of the canonical Serbian Orthodox Church brought about the demise of the Democratic Party of Socialists of President Milo Đukanović and its loss of the majority in the parliamentary election for the first time after the country gained independence in 2006. This was preceded by an attempt to take away the acquired rights and property enjoyed by the Serbian Orthodox Church in Montenegro, as well as the application of a series of repressive measures, including violence against the archbishops, priests, monks and believers, which was justified even with reference to the COVID-19 pandemic (Mirović 2020).

[4]　The traditional churches and religious communities in Serbia are: the Serbian Orthodox Church, the Islamic community, the Catholic Church, the Slovak Evangelical a.c. Church, the Jewish community, the Christian Reformed Church and the Evangelical Christian Church a.c., and for each of them confessional instruction classes are organized in the environments and schools where parents specifically want their children to attend those classes.

<sup>5</sup> According to the data of the Serbian Orthodox Church, in Serbia in 2021 as many as 2010 religious instruction teachers were been employed: 1602 Orthodox, 247 Islamic, 131 Roman Catholic and 30 instructors from the reformation churches. Their classes were attended by about 450,000 school students, whereas 315,000 of them attended civic education classes (Maričić 2021).

<sup>6</sup> After the introduction of religious instruction as a subject, textbooks were published by the same publishing house for this subject for the first grade of primary school in Croatian, Hungarian (Roman Catholic), Bosniak (Ilmuddin) and Albanian (Ilmuddini), Ruthenian and Ukrainian (Greek Catholic), the Roma language (Orthodox) and Romanian (Orthodox). For the second grade of primary school they were published in Croatian and Hungarian (Roman Catholics) and Ruthenian (Greek Catholic). For the third, fourth, fifth and sixth grades of primary school they were published in Croatian and Hungarian (Roman Catholics). For the first and second grades of secondary school they were published in Albanian (Ilmuddini) and Bosniak (Ilmuddin). For the first grade of secondary school they were published in Croatian and Hungarian for students of Roman Catholic confessional studies.

<sup>7</sup> Although it has been determined that only the confessional model of religious instruction is in line with the Constitution of Germany, it is clear that nowadays it is subject to criticism, primarily from the aspect of advocation for the interculturalization of religious instruction (Zonne-Gätjens 2022). Unlike Germany, Serbia is not a migrant country, but its religious, confessional and ethnic diversity would also impose the need for students to become familiar with religious differences.

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
