# Peer review of "Confessional Instruction or Religious Education: Attitudes of Female Students at the Teacher Education Faculties in Serbia"

_religions, doi:10.3390/rel14020160_

Round 1
Reviewer 1 Report
This is an interesting focus and survey with potential but I suggest the following:
1. Provide a clearer, more historical context to the role and status of religion in schools in Serbia (especially in light of the wars of the 1990s and its link to what to many Bosniaks in particular is deemed genocide).
2. Provide some examples of the textbooks used in schools and evaluate how and whether it is open to interreligious and nonreligious positions. Where are the key biases or absent voices? Refer to statements by various religious leaders across the faith spectrums for their views on the curriculum.
3. The current version of the Introduction is not particularly helpful and states a number of unhelpful or inaccurate statements about religion broadly or about specific religions.
a. Do all religions intend to universalize (pg 2, line 67) or only certain religions with strong missionary elements?
b. Stating Christ proclaimed a new monotheistic religion (p1, line 28) is a blatant example of supersessionism—decried in most mainstream Christian circles today, especially In light of Jewish-Christian dialogue and relations. Jesus was Jewish of course and what became Christianity is rooted in Judaism.
c. Islam is the proper name of the religion referred to in line 38 of pg 1
d. The opening line is distracting. I would focus on your survey and giving the non-specialist and those unfamiliar with Serbia and religious education a clearer, more comprehensive background of religion in Serbia and in the schools.
4. Footnote 3 is important and needs a clearer statement, especially in light of the genocide in Srebrenica, among other places. As you know, there are a lot of books in this area and views are sometimes radical, but I suggest this fairly balanced account, by Peter Lippman, Surviving the Peace: The Struggle for Postwar Recovery in Bosnia-Herzegovina (Nashville, TN: Vanderbilt University Press, 2019)-- that while contending Bosniaks suffered the most, does not deny that there were also innocent Serbians who unjustly were killed and/or tortured. See other works (with a sharper sense of blaming Serbian leadership) but still are valuable and include: The War is Dead, Long Live the War by Ed Vulliamy and Surviving the Bosnian Genocide: The Women of Srebrenica Speak by Selma Leydesdorff. The comment on Russia’s invasion and the claims of torture and genocide committed by Russian soldiers with the support of the Russian leadership could also use more clarity here (if it can fit with your overall aims).
Author Response
Thank you very much for your marks. We are sending to you new version of the article.
Reviewer 2 Report
The research is interesting. At least I miss a comparison with similar surveys in other countries. The introduction is too long and too general.
In the conclusion, you are too quick to assume that religious education is at odds with science and modern goals of education. Yet it is not necessarily the case that denominational instruction leads to a lack of understanding of other religions. Good examples of its positive effect can be found in Germany and Austria.
It would also be interesting to see, despite the small number of male students, if there is a difference between the sexes in terms of acceptance of religious education in school.
It would be good if it was pointed out in the keywords that this is just an example of religious education only in Serbia!
Author Response
Thank you very much for your help. We improved our paper according to your constructive suggestions and we are sending to you new version of the article.
